# Flexural Strengthening of RC Structures with TRC—Experimental Observations, Design Approach and Application

**Silke Scheerer [1,\*], Robert Zobel [2], Egbert Müller [1], Tilo Senckpiel-Peters [1], Angela Schmidt [1] and Manfred Curbach [1]**

[1]  Institute of Concrete Structures, TU Dresden, 01069 Dresden, Germany;
    Egbert.mueller@tu-dresden.de (E.M.); Tilo.senckpiel-peters@tu-dresden.de (T.S.-P.);
    Angela.schmidt@tu-dresden.de (A.S.); Manfred.curbach@tu-dresden.de (M.C.)
[2]  IBB Ingenieurbüro Baustatik Bautechnik, TU Dresden, 01069 Dresden, Germany; Zobel@ibb-wilhelm.de
\*  Correspondence: Silke.scheerer@tu-dresden.de; Tel.: +49-351-463-36527

**Abstract:** Today, the need for structural strengthening is more important than ever. Flexural strengthening with textile reinforced concrete (TRC) is a recommendable addition to already proven methods. In order to use this strengthening method in construction practice, a design model is required. This article gives a brief overview of the basic behavior of reinforced concrete slabs strengthened with TRC in bending tests as already observed by various researchers. Based on this, a design model was developed, which is presented in the main part of the paper. In addition to the model, its assumptions and limits are discussed. The paper is supplemented by selected application examples to show the possibilities of the described strengthening method. Finally, the article will give an outlook on open questions and current research.

**Keywords:** textile reinforced concrete (TRC); strengthening; bending; model; design; practical application

---

## 1. Introduction—Strengthening of Concrete Buildings—Why and How

Particularly in the past decades, the world population has grown rapidly and will continue to do so according to unanimous forecasts. Buildings are essential for mankind, but neither the natural resources nor the available room or the costs allow us to cover our needs by new buildings alone. Buildings, roads, and bridges must be carefully maintained, renovated, or reinforced if their load-bearing capacity is no longer given or if there are deficiencies in their serviceability.

Reinforced concrete has been the world's most widely used composite building material for more than a hundred years. When properly dimensioned and constructed, it is very efficient and durable. Two things in particular can be problematic:

1.  Corrosion of the steel reinforcement; it can occur because of carbonation of concrete, the choice of unfavorable materials, insufficient concrete cover, or too wide cracks.
2.  Insufficient load-bearing capacity; this mostly results from the loads which have steadily increased over the decades and which must be taken into account according to the standard. One example is the increased axle loads of trucks. About 100 years ago, the total weight of commercial vehicles was about 10 tons. Today, gigaliners with total weights of up to 60 tons are under discussion worldwide; however, their traffic-legal approval is the responsibility of national authorities.

Various methods exist for the rehabilitation and strengthening of plain, reinforced, and prestressed concrete structures, see e.g., References [1,2]. The most important are briefly mentioned:

- Shotcrete with steel reinforcement, e.g., References [3–5],
- Glued or notched steel lamellas, e.g., References [6–9],
- Glued or notched lamellas/sheets made of fibre-reinforced plastics (abbreviated: FRP), e.g., References [7–16], or
- Supplementation of additional components such as external tendons, e.g., References [17,18].

Another method that has become increasingly established over the past 20 years is strengthening with textile reinforced concrete (abbreviated: TRC; also known as textile reinforced mortar, abbreviated: TRM, or fibre-reinforced cementitious matrix (abbreviated: FRCM) composites) [19–24]. Similar to shotcrete, TRC is—from the material's side—very compatible with the steel reinforced concrete of the primary building element. Because the fibres used in TRC can bear higher tensile stresses than steel and do not corrode, considerably smaller layer thicknesses can be realized compared to conventional shotcrete. Compared to steel lamellas, the requirements for corrosion protection are lower. Like FRP lamellas, TRC is a relatively light building material, which can be processed by hand. The concrete cover around the fibre reinforcement in TRC is advantageous at elevated temperatures—for the textiles themselves as well as for the steel reinforcement in the basic component.

A major disadvantage of textile reinforced concrete is—with a few exceptions, e.g., References [23,25–27]—the lack of general building authority approvals, guidelines, and standards. To date, not all details of the load-bearing behavior of TRC have been clarified. In addition, there is a constantly growing number of further and new developments in the field of fibre reinforcement. Often these reinforcements show a similar load-bearing behavior. In detail, however, there can be differences that must be taken into account when dimensioning and applying textile reinforced concrete.

TRC can be used to increase the bending, torsional, longitudinal, and shear load-bearing capacity and to improve serviceability and functionality, e.g., References [22,24]. In this article, the focus lies on a model to calculate the increase of the flexural load-carrying capacity due to an additional TRC layer. On the one hand, compared to the other TRC strengthening variants, this has been very well researched. On the other hand, in our experience, there is a great need for flexural strengthening measures in building practice compared to the other possibilities.

## 2. Research on Flexural Strengthening with Textile Reinforced Concrete

First research with textile fabrics made of technical endless fibres embedded in mortar and fine-grained concrete respectively has been performed since the early 1990s, e.g., References [28–30]. At our institute, we carried out first own tests on steel reinforced (abbreviated: RC) concrete plates strengthened with an additional TRC layer in the bending tensile zone in 1997 as part of a feasibility study [31,32]. The objective was to demonstrate the potential of textile reinforcements to increase the flexural load-carrying capacity of RC slabs in principle. The three plate specimens were made of normal strength concrete B25 (acc. to DIN 1048 [33], this corresponds nearly to the current concrete class C20/25 acc. to DIN EN 206 [34]), and were reinforced with 5 steel bars with a diameter of 8 mm (Figure 1a). Two of the plates were strengthened with 6 textile layers embedded in fine-grained mortar—at that time still using ISBOTON as adhesion promoter. The used multiaxial grids were made of alkali-resistant (AR) glass filaments (Figure 1b). The tensile strength of the glass fibres ranged between 1200 and 1400 MPa, the ultimate strain was 20%. The strengthening layer was applied to the whole underside of the specimen. In all following examinations, the support areas remained unstrengthened. The plates were studied in 3-point bending tests. The load was applied path controlled; deformations were recorded by strain gauges and linear variable differential transformers (LVDT). Figure 1c displays load-middle deflection curves of the tests. The blue line shows all characteristics of an RC slab subjected to bending. At a deflection of approximately 1 mm, a first crack was formed, followed by a short phase of formation of further cracks. After that, a steady, almost linear increase in deflection could be detected (state II). At a deflection of 8.5 mm, observed at a load of 20 kN, the steel reinforcement reached its yield strength. From now on, no further load increase was possible.

On account of the steel reinforcement's plastic deformation capability, only the deflection increases until a final bending failure occurred.

The two strengthened plates (red and green in Figure 1c) showed a similar behavior to each other, but a different behavior compared with plate 1. The first crack opened at circa 16 kN, and the associated load level was therefore twice as high as at plate 1. With the increase of the load (state II), the deflection then increased again linearly, but—compared to plate 1—at a higher load level resulting from the additional textile reinforcement. When a load of approx. 35 kN and a deformation of approx. 12 mm were reached, the textile reinforcement teared and the force dropped down to the level given by the yield point of the steel bars. In summary, the additional reinforcement with 6 layers of AR glass textile increased the load by 69% (average value of the two tests). Upon reaching the maximum load the deflection increased by approx. 30%. Still, at the same load level, the deflection of the reinforced components was less.

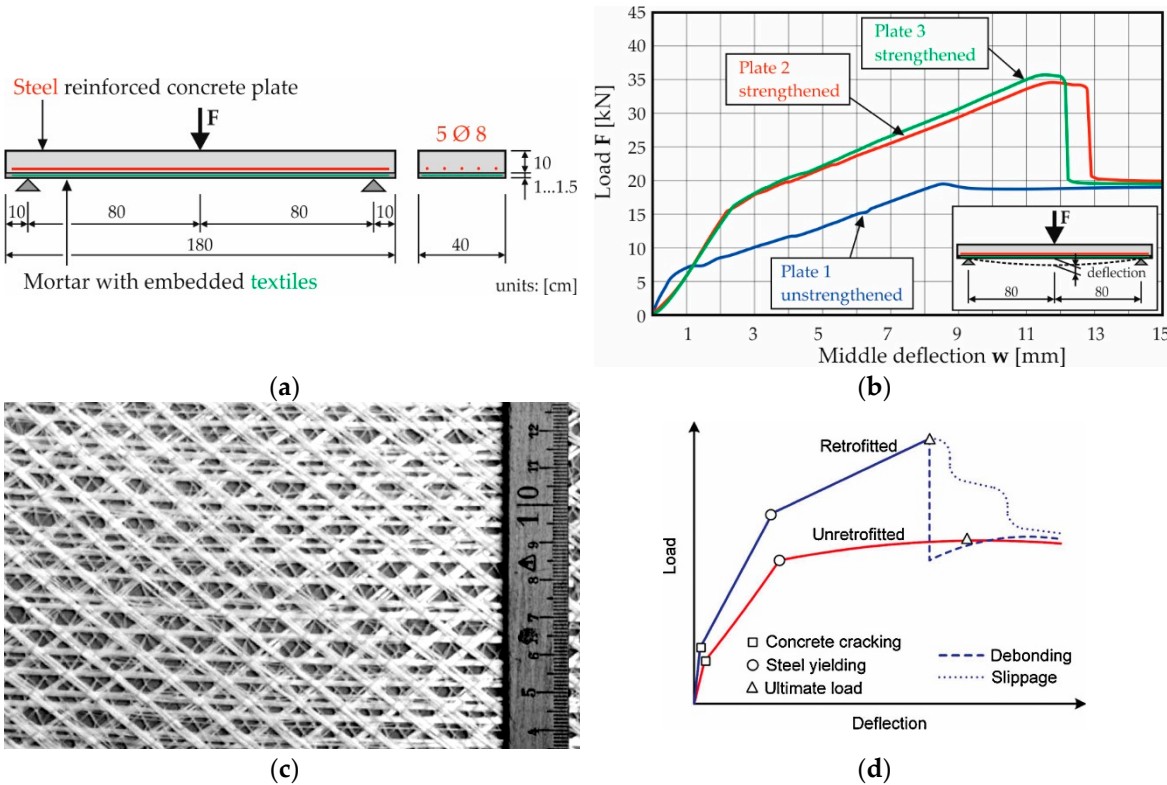

**Figure 1.** First test on reinforced concrete (RC) slabs strengthened with textile reinforced concrete (TRC) in Germany—(**a**) test set-up acc. to Reference, (**b**) used textile made of alkali-resistant (AR) glass, produced by the Institute for Textile and Clothing Technology of TU Dresden, (**c**) test results ((**a**–**c**) reproduced and modified with permission from [31], Institute of Concrete Structures, TU Dresden, 1997) and (**d**) principle behavior of TRC-strengthened RC plates or beams (reproduced from Koutas et al. [22] on the basis of the Creative Commons Attribution license (http://creativecommons.org/licenses/by/4.0/) under which this article open access was published, ASCE Library, 2019).

These first tests already demonstrated the basic phenomena of TRC-strengthened RC slabs or beams subjected to bending:

- Initial cracking at a higher load level,
- Reduction of deflection (at same reference load),
- Increase of the bearable load until failure.

In addition, in state II, the stiffness of TRC-strengthened components is often higher than that of unstrengthened RC elements. This is caused by the higher component's thickness and—often

the decisive factor—by the increased elongation stiffness in the tensile zone (described in detail, for example, in Reference [35]).

Today a large number of research projects and publications on flexural strengthening of RC structures with TRC are known, in which comparable phenomena have been observed. Exemplary for all, we like to recommend the publication of Koutas et al. [22] and Carloni et al. [23]. The authors collected, studied, analyzed, and evaluated numerous research works from all over the world on the subject "strengthening of concrete structures with TRM" (TRC). In addition to the often considered uniaxial case, studies on two-way slabs are listed as well as experiments at elevated temperature; furthermore, various research projects on the behavior of TRC-strengthened components under impact loading are known (see for example Reference [36–39]). In References [22,23] the described basic phenomena are confirmed, based on an extensive evaluation of known experiments worldwide. Figure 1d summarizes the component's behavior in a schematic sketch by Koutas et al. [22]. The specific shape of such a load-deformation curve depends on the type and the quantity of the textile reinforcement, and, of course, on the characteristic of the RC element to be strengthened. To add a last characteristic: In general, textile reinforcements with yarn distances in the range of several millimeters up to a few centimeters cause a finer crack pattern in the bending tensile zone compared to steel reinforced concrete.

The type of failure is of particular interest for the design or practical application of a flexural strengthening with TRC. In general, the load-bearing capacity of RC components subjected to bending is reached when either the concrete fails under compression or the steel reinforcement ruptures under tension. In addition, there are failure mechanisms associated with shear (especially for beams) or detailed problems such as failure due to insufficient anchoring of the bending tensile reinforcement. If the tensile strength of the reinforcement limits the bearing capacity of a component, the application of a textile concrete layer in the bending zone can increase the bearing capacity. In doing so, it must be ensured that the "old" concrete in the pressure zone has sufficient load-bearing capacity. Otherwise, the failure mode may change. It should be noted in particular that the addition of a TRC layer with high load-bearing capacity, on the one hand, could defer crack formation in the bending tensile zone and cause a sudden concrete compression failure (see Section 3.3.3), and on the other hand (compare Reference [21] and Section 5), can cause also shear failure. Furthermore, failure mechanisms may occur which are not known from reinforced concrete construction. The following scenarios must be considered, compare e.g., References [22,24,35,40]:

- Similar to RC and steel reinforcement—exceeding the tensile strength of the textile reinforcement; this failure is indicated by increasing crack formation and deflection and is the quasi "wanted" failure form; the load-deformation behavior is essentially determined by the mechanical properties of the textile reinforcement (tensile strength, modulus of elasticity).
- Forms of failure due to the transfer of forces from the strengthening layer into the reinforced concrete base body:

  ○ Failure inside the old concrete (that means the concrete of the structural member to be strengthened fails first, often near the joint between old and new concrete; it is known also from other strengthening methods),
  ○ Delamination in the joint between old concrete and TRC layer in the end anchorage area or at opening cracks,
  ○ Delamination or debonding within the TRC layer (usually in the plane of the most stressed textile grid),
  ○ Extraction of the reinforcement from the matrix ("slippage" of the fibres through the mortar in Figure 1d).

The last two failure variants are primarily dependent on the inner bond within the yarns and on the bond between the yarn's edge filaments and the surrounding concrete matrix. The size

and impregnation used have a major influence. Even though some approaches have already been developed to explain these phenomena, the topic of failure forms and bonding is still the subject of ongoing research.

In summary, much knowledge has been generated worldwide on the subject of bending retrofitting with TRC. In addition to many similarities, these results are often not directly comparable, since the known studies differ in many details. The main differences are the type of textile reinforcement, its characteristic material properties and amount, the properties of the strengthened basic RC members (geometry, material, reinforcement), and the test set-up (generated distribution of inner forces). Nevertheless, bending strengthening layers can in principle be calculated with relatively uncomplicated models. One possibility is presented and discussed in the following chapter.

## 3. Design Concept for Flexural Strengthening of Reinforced Concrete (RC) Components with Textile Reinforced Concrete (TRC)

### 3.1. General Information

There are standards on design rules and models for RC structures, e.g., in Reference [41]. In case of bending, the calculation can be done by an iterative process. A similar procedure was also chosen for the calculation of the flexural strengthening with TRC. In accordance with the design method for steel reinforced concrete, the following assumptions and simplifications are defined (compare e.g., References [41,42] and also References [23,36]):

- Cross sections remain plane (Bernoulli hypothesis).
- Strain compatibility between reinforcement and concrete is assumed.
- The concrete's tensile strength is ignored; all tensile forces are taken up by steel and textile reinforcements.
- Rigid bond between steel, concrete, and textile reinforcement may be assumed.
- The design is carried out at the ultimate limit state (ULS), i.e. at least one material (concrete, reinforcing steel, or textile reinforcement) reaches the ultimate strain.

A first model was presented by Bösche in Reference [43] in 2007. Meanwhile, it was further developed [42,44–47] and modified due to new knowledge and research findings for the later used materials. Furthermore, a similar design proposal is presented in Carloni et al. [23]. The main difference lies in the choice of the reference plane for the formation of the internal equilibrium. Compared to Reference [23], the model presented in the following has the advantage that the reference horizon is fixed by the geometry and the structural design of the component (position of the reinforcements) and does not have to be determined by stress or strain distribution.

### 3.2. Material Models

A standard material model for concrete was chosen [41,48,49], Figure 2a, and can be simplified into two sections—a parabolic section until the strain $\varepsilon_{c2}$, and a linear horizontal branch until the ultimate strain $\varepsilon_{cu2}$ is reached. The values $\varepsilon_{c2}$ and $\varepsilon_{cu2}$ depend on the concrete class of the structural member that has to be strengthened.

The material model for the steel reinforcement is also taken from Reference [41,48,49], Figure 2b. It can be described as a bilinear curve, either with or without a strength increase (hardening) after reaching the yield strength.

Depending on the fibre material, there are differences in the stress–strain behavior of textile grids. The design model used here was primarily developed on the basis of tensile tests with carbon fibre reinforcements (e.g., [50]). When using material with different characteristics, the here mentioned recommendations may have to be modified. In Figure 3, different possible stress–strain relationships for carbon reinforcement are shown. The variant 1 (Figure 3a, used in Reference [25,46]) derived from material behavior observed in tension tests in former years where the investigated textile grids made

of AR glass or carbon have shown a less stiff stress–strain behavior at lower load levels, e.g., due to undulation of yarns after manufacturing [45–47]. The meanwhile available carbon reinforcements show a nearly linear material behavior which led to a modification of the material model, variant 2 in Figure 3b (compare e.g., [47]) and variant 3 in Figure 3c, respectively. In Carloni et al. [23] and in Curbach et al. [47], a stress–strain curve with constant elastic modulus as in variant 3c is recommended. The different variants will be discussed in Section 3.4.1.

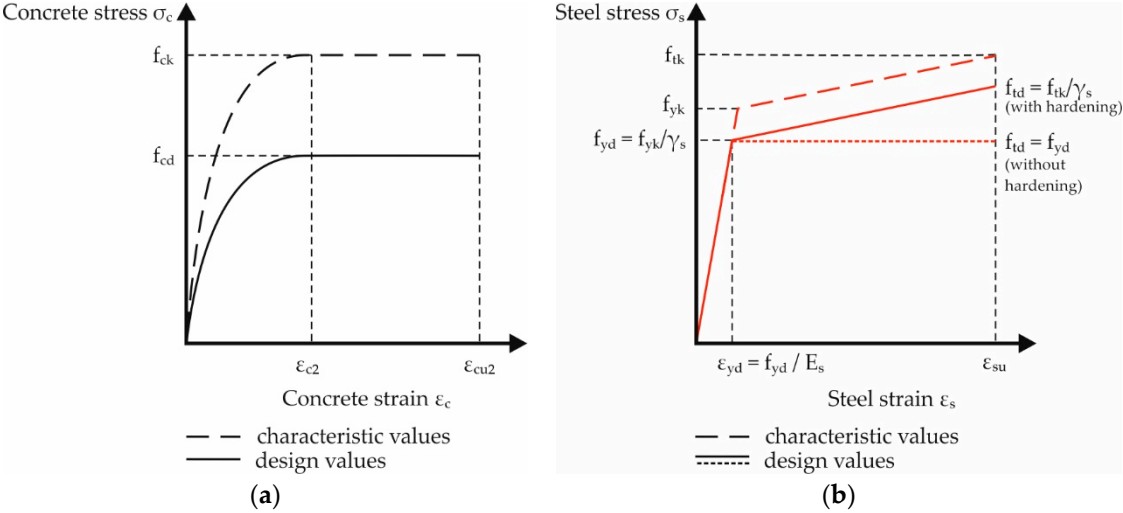

**Figure 2.** Stress-strain relationships of the reinforced concrete (RC) components according to European code: (**a**) parabola-rectangle diagram for concrete under compression; (**b**) bilinear diagrams for reinforcing steel (tension and compression).

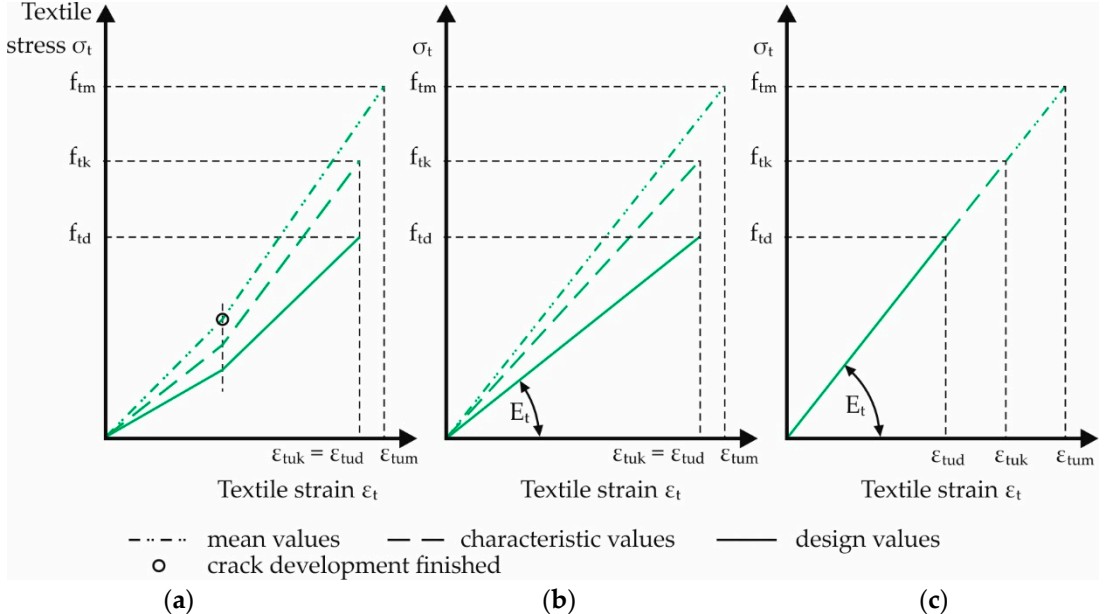

**Figure 3.** Possible stress-strain relationships for carbon reinforcements: (**a**) variant 1: for textiles with weaker behavior at low stress levels, e.g., because of undulated yarns; (**b**) variant 2: for textiles with linear stress–strain behavior (variable modulus of elasticity); (**c**) variant 3: for materials as in (**b**), constant modulus of elasticity.

*3.3. Calculation Model*

3.3.1. Iteration Process

Similar to RC, the design model is based on the equilibrium of internal and external forces. As is well known, this equilibrium cannot be solved in a closed way, one or more iterations are necessary. The iteration process is described e.g., in References [43,46,47]. In this paper, the basic formulas will be given for a cross section with forces, stresses, and strains according to Figure 4 (compare e.g., [46,47]). The definitions of the symbols are summarized in Table 1.

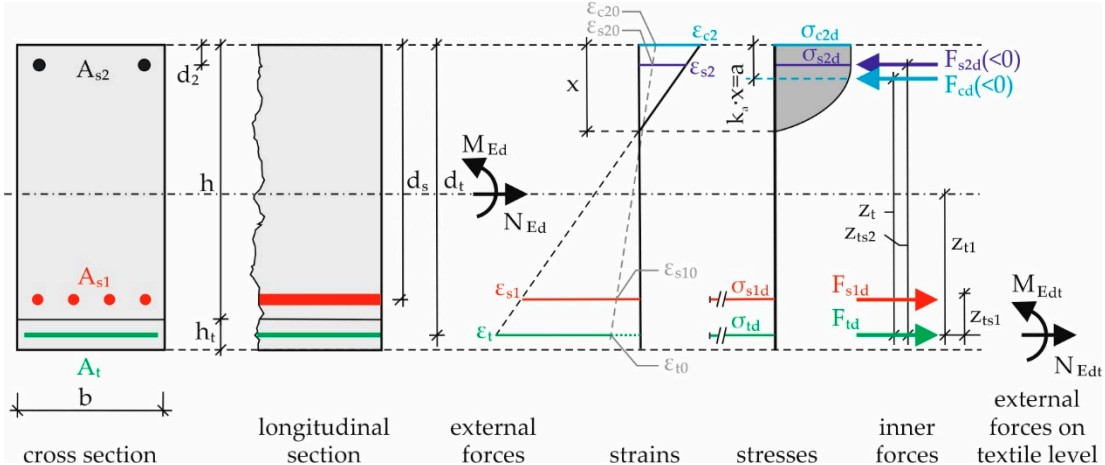

**Figure 4.** Principle of strengthened reinforced concrete (RC) cross section with outer and inner forces, strains and stresses.

**Table 1.** Definition of symbols.

| Symbol | Definition |
|:---:|:---:|
| **Indices** | |
| E | external |
| c | concrete |
| s | steel |
| t | textile |
| d | design value |
| 1 | tensile stress area |
| 2 | compressive stress area |
| 0 | preload condition |
| **Forces and Moments** | |
| F | force |
| N | normal force |
| M | bending moment |
| **Geometrical Values** | |
| h | height |
| d | effective depth |
| b | width |
| A | reinforcement area |
| a | distance of compression force to top layer |
| x | neutral axis depth |
| z | inner lever arm |
| $k_a$ | coefficient for distance a |
| $\alpha_R$ | block coefficient |
| **Stresses and Strains** | |
| σ | stress |
| ε | strain |

The maximum resistance of the cross section is reached when at least one of the three materials reaches its ultimate capacity. The goal is to fully utilize the textile reinforcement. This is therefore the first limiting case: The textile reinforcement achieves its ultimate strain ($\varepsilon_{tu}$) or ultimate strength. The second possibility is a concrete failure in the compression zone ($\varepsilon_c = \varepsilon_{cu2}$). Last but not least, the steel reinforcement can fail as well, when the steel strain reaches $\varepsilon_{su}$, but this scenario would require an extremely large pre-deformation of the existing component prior to the strengthening measure (see "consideration of a preload", Section 3.3.3), and would apply only to exceptional cases. All other types of failure, e.g., debonding or anchorage failure, are to be excluded by separate proofs or by complying with design regulations analogous to steel reinforced concrete construction.

The first equation for the iteration is the equilibrium for horizontal forces. Irrespective of the graphic representation in Figure 4, compressive forces are introduced in equilibrium conditions under consideration of the sign (negative!).

$$N_{Ed} = F_{s1d} + F_{td} + F_{s2d} + F_{cd} \tag{1}$$

In Equation (2), the moment's equilibrium is formed in the centre of gravity of the textile reinforcement.

$$M_{Ed} - N_{Ed} \cdot z_{t1} = -F_{s2d} \cdot z_{ts2} - F_{cd} \cdot z_t - F_{s1d} \cdot z_{ts1} \tag{2}$$

In the Equations (1) and (2), there are four unknown forces ($F_{s1d}$, $F_{td}$, $F_{s2d}$, $F_{cd}$) and one unknown geometrical value $z_t$. Overall, there are two equations but five unknown variables. To solve this mathematical problem, more equations are necessary. For this reason, the strains will be brought into relation, according to the Bernoulli hypothesis.

$$\varepsilon_{s1} = \varepsilon_{c2} + (\varepsilon_t + \varepsilon_{t0} - \varepsilon_{c2}) \cdot \frac{d_s}{d_t} \tag{3}$$

$$\varepsilon_{s2} = \varepsilon_{c2} + (\varepsilon_{s1} - \varepsilon_{c2}) \cdot \frac{d_2}{d_s} \tag{4}$$

$$\varepsilon_{t0} = \varepsilon_{s10} + (\varepsilon_{s10} - \varepsilon_{c20}) \cdot \frac{z_{ts1}}{d_s} \tag{5}$$

For iteration, a first strain distribution must be estimated. Then, the position of the neutral axis x, including all coefficients such as $k_a$ and $\alpha_r$, and the compression force $F_{cd}$ can be calculated. With the help of the geometric values of effective depth of the textile $d_t$ and the centre of gravity of the resulting concrete compressive force a, the inner lever arm $z_t$ can be determined. Due to the given amount of steel reinforcement and the according stress–strain relationship, the steel stresses $\varepsilon_{s1}$ and $\varepsilon_{s2}$ can be calculated. Now the equilibrium of moments can be checked. If it is fulfilled, Equation (1) can be used to determine the necessary textile reinforcement according to the material characteristics of the grid. If it is not fulfilled, the assumed strain distribution has to be modified and the calculation must be repeated.

### 3.3.2. Design Tables for Calculation

The iteration process described before can be time-consuming. Therefore, dimensionless design tables can be developed, e.g., References [46,47]. Analogous to RC calculation tables, specific application limits must be taken into account, e.g., the concrete strength class and the kind of steel reinforcement. In the case of flexural strengthening with TRC, the characteristic material properties of the textile reinforcement (stress–strain relation) must be considered as well. That means, for every kind of textile, special design tables have to be created. For textiles that are frequently used, however, it is worth the effort to generate such tables because overall the handling is much more efficient compared to the iteration.

For rectangular cross sections made of normal strength concrete and reinforcing steel acc. to EC2 [41,48,49] and carbon reinforcement acc. to Reference [25], (ed. 2016, characteristic ultimate strain $\varepsilon_{tuk}$ = 0.0075), design tables were already developed (Frenzel [46], stress–strain relation according to Figure 3a; Zobel [47] has used the more linear stress–strain relation of modern textiles acc. to Figure 3b, as well as higher design values for the tensile strength of carbon). In Table 2, a stress–strain distribution with constant Young's modulus according to Figure 3c was used, recommended by the authors and Carloni et al. [23]. Table 2 does not include reductions in regard to durability, temperature, and permanent load when determining the design value of the carbon material strength; for discussion, see Section 3.4. In addition, in Table 2, strain hardening of the steel reinforcement was not considered (horizontal branch of σ-ε curve after reaching the yield strain); the preload $\varepsilon_{t0}$ is assumed to be zero.

**Table 2.** Dimensionless design table for rectangular reinforced concrete RC member flexurally strengthened with textile reinforced concrete (TRC [1]).

| $\mu_t$ | $\omega_t$ | $\xi_t = x/d_t$ | $\zeta_t = z_t/d_t$ | $\varepsilon_{c2}$ (‰) | $\varepsilon_t$ (‰) | $\sigma_t$ (N/mm$^2$) | |
|---|---|---|---|---|---|---|---|
| 0.01 | 0.0102 | 0.059 | 0.980 | −0.37 | 5.95 | 1291.67 | |
| 0.02 | 0.0206 | 0.084 | 0.972 | −0.54 | 5.95 | 1291.67 | |
| 0.03 | 0.0311 | 0.103 | 0.965 | −0.68 | 5.95 | 1291.67 | Rupture of textile reinforcement |
| 0.04 | 0.0417 | 0.120 | 0.959 | −0.81 | 5.95 | 1291.67 | |
| 0.25 | 0.2947 | 0.366 | 0.848 | −3.43 | 5.95 | 1291.67 | |
| 0.26 | 0.3091 | 0.382 | 0.841 | −3.50 | 5.67 | 1230.68 | |
| 0.27 | 0.3239 | 0.400 | 0.834 | −3.50 | 5.25 | 1139.59 | Failure of concrete |
| 0.34 | 0.4391 | 0.542 | 0.774 | −3.50 | 2.95 | 641.40 | |
| 0.35 | 0.4576 | 0.565 | 0.765 | −3.50 | 2.69 | 584.60 | |

[1] stress–strain relationship acc. to Figure 3c, no preload ($\varepsilon_{t0}$ = 0).

The handling is easy. First, it is assumed that the steel strains $\varepsilon_{s1}$ and $\varepsilon_{s2}$ reached or exceeded the yielding point. With Equation (6), the design bending moment $\mu_t$ can be calculated:

$$\mu_t = \frac{M_{Edt} + F_{s1d} \cdot z_{ts1} + F_{s2d} \cdot z_{ts2}}{b \cdot f_{cd} \cdot d_t^2} \tag{6}$$

Forces and moment can be calculated as follows:

$$F_{s1d} = A_{s1} \cdot f_{yd}; \ F_{s2d} = A_{s2} \cdot f_{yd}; \ M_{Edt} = M_{Ed} - N_{Ed} \cdot z_{t1}$$

The textile reinforcement ratio $\omega_t$ and the strains $\varepsilon_t$ and $\varepsilon_{c2}$ can be taken from Table 2. In the next step, the previous assumption for the strains $\varepsilon_{s1}$ and $\varepsilon_{s2}$ (yield range) must be checked with Equations (7) and (8).

$$\varepsilon_{s1} = \frac{\varepsilon_t - \varepsilon_{c2}}{d_t} \cdot d_s + \varepsilon_{c2} > \varepsilon_{yd1} \tag{7}$$

$$\varepsilon_{s2} = \left| \varepsilon_t - \frac{\varepsilon_t - \varepsilon_{c2}}{d_t} \cdot (d_t - d_2) \right| > \varepsilon_{yd2} \tag{8}$$

(a) If Equations (7) and (8) are fulfilled, the textile reinforcement area $A_t$ can be determined on the basis of the following formula (9):

$$A_t = \frac{1}{\sigma_t} \cdot (\omega_t \cdot b \cdot f_{cd} \cdot d_t + N_{Ed} - F_{s1d} - F_{s2d}) \tag{9}$$

(b) If the yield strains are not reached, $\mu_t$ must be recalculated (Equation (6)), taking into consideration the strains calculated by (7) and (8), as listed below:

$$F_{s1d} = A_{s1} \cdot f_{yd}, \ \text{if} \ \varepsilon_{s1} \geq \varepsilon_{yd1}; \ F_{s2d} = -A_{s2} \cdot f_{yd}, \ \text{if} \ |\varepsilon_{s2}| \geq \varepsilon_{yd2}$$

$$F_{s1d} = E_{s1} \cdot \varepsilon_{s1}, \ \text{if} \ \varepsilon_{s1} < \varepsilon_{yd1}; \ F_{s2d} = E_{s2} \cdot \varepsilon_{yd}, \ \text{if} \ |\varepsilon_{s2}| < \varepsilon_{yd2}$$

Now, the textile reinforcement ratio $\omega_t$ and the strains $\varepsilon_t$ and $\varepsilon_{c2}$ can be selected again from Table 2. In the next step, the "new" defined strains $\varepsilon_{s1;new}$ and $\varepsilon_{s2;new}$ must be checked with formulas (10) and (11) as well.

$$\varepsilon_{s1;new} = \frac{\varepsilon_t - \varepsilon_{c2}}{d_t} \cdot d_s + \varepsilon_{c2} \approx \varepsilon_{s1} \tag{10}$$

$$\varepsilon_{s2;new} = \varepsilon_t - \frac{\varepsilon_t - \varepsilon_{c2}}{d_t} \cdot (d_t - d_2) \approx \varepsilon_{s2} \tag{11}$$

If the values $\varepsilon_{s1}$ and $\varepsilon_{s1}$ do not match, the procedure must be repeated from point (b) until the Equations (10) and (11) are fulfilled.

Finally, Equation (9) can be used to determine the required textile reinforcement area $A_t$.

### 3.3.3. Consideration of a Preload $\varepsilon_{t0}$

In some publications, the influence of a preload applied to the basic component before it was strengthened have been already addressed, e.g., References [22,51] with a focus on experimental results or [46,47] with regard to calculation. The thesis is that the imprinted strain state of the initial component is of particular interest for the design. Therefore, the preload influence on the design of a flexural strengthening TRC layer, already discussed in Reference [46] and further examined in Reference [47], shall be summarized.

Figure 5 displays the relation between the design bending moment $\mu_t$ and a pre-deformation $\varepsilon_{t0}$. $\varepsilon_{t0}$ is a virtual size and can be determined from strain distribution of the pre-deformed component and by an assumption of the position of the textile grid in the reinforcing layer (compare Figure 4), taking a plane cross section into account (Bernoulli). Now, $\varepsilon_{t0}$ was used as an input value for further strain analysis. In the diagram, there are two different colored areas. In the grey one, the ultimate strain of the concrete, and in the green one, the ultimate strain of the carbon reinforcement is reached in ULS. The corresponding other strain value varies from its maximum and minimum respectively to zero (compare the different colored lines in the diagram). In the green area, tensile failure of the textile occurs; the carbon reinforcement is fully utilized. In the grey zone, concrete failure will happen. The dark blue line is the limit, where concrete and carbon reinforcement fail at the same time. It can be concluded [46,47] that for the same design bending moment $\mu_t$ and increasing pre-deformation $\varepsilon_{t0}$, the utilization of the concrete compression zone increases. Strain distribution, type of failure, and required textile reinforcement $A_t$ are therefore dependent on the pre-stress condition of the unstrengthened RC cross section.

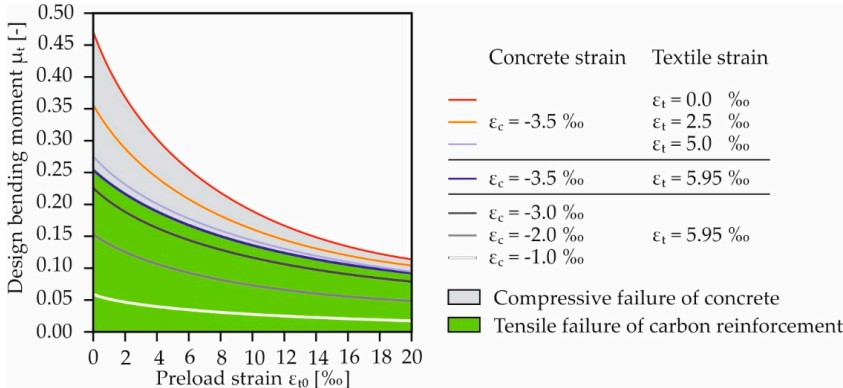

**Figure 5.** Influence of a pre-deformation on the strain plane and failure mode of a retrofitted reinforced concrete (RC) member as a function of the design bending moment; published by Zobel in Reference [47], modified (diagram is based on a stress-strain distribution of the carbon reinforcement acc. to Figure 3c and Reference [25]).

*3.4. Special Aspects to be Taken into Account When Applying the Design Proposal*

Similar to reinforced concrete construction, flexural design based on the equilibrium of internal and external forces is a reliable method. The method presented here has been tested by means of various experimental investigations. Altogether, satisfactory results were achieved. Nevertheless, there are still aspects to be discussed. These include the assumption of a stress–strain relationship for the textile reinforcement as well as the questions, which partial safety factors or which distribution function are to be used for the scattering material properties. Furthermore, not all failure phenomena observed by different researchers [22,23] so far have yet been conclusively explained. In this sense, the following sections are to be understood as a basis for discussion.

3.4.1. Influence of the Used Stress–Strain Relationship for the Textile Reinforcement

The design table presented in Section 3.3.2 uses the stress–strain curve for a carbon fibre strand shown in Figure 3b. For the reliable determination of the component's load-bearing capacity, this curve is reduced in certain values by a partial safety factor. For the material design curve currently in use, only the strength is reduced. However, this is also accompanied by a reduction in the modulus of elasticity. This corresponds to the procedure, which is also used when applying the stress–strain curve in concrete design. On the other hand, in Reference [23] or Reference [47] a further variant is presented (Figure 3c). In this variant, the strength and the elongation are reduced to the same extent, so that the modulus of elasticity does not change. This corresponds to the procedure which is also used in the design of reinforcing steel—at least for the linearly elastic, first section of the reinforcing steel stress–strain line until the yield point is reached. Here, the characteristic value of the yield strength is first reduced by the partial safety factor, after that the corresponding elongation is determined by division with the modulus of elasticity of the reinforcing steel. In the variant from Figure 3c for the carbon textile, the procedure is the same. First, the breaking stress is determined by means of single fibre tensile tests [52] or uniaxial tensile tests, e.g., according to Reference [50], and then the elongation at failure is determined by division of the strength by the modulus of elasticity. The reduction of the strength and elongation values takes place to the same extent, so that the modulus of elasticity remains the same before and after the reduction.

What effects does this changed procedure have on the load-bearing capacity design, since the value of the elongation at fracture plays an important role as one of the two limit strains in the iteration process for cross sectional design? With variant from Figure 3c, a stiffer bending component behavior is obtained by reaching the maximum textile stress earlier with smaller elongation at failure, since the elongation at failure is also reduced. This results in a faster utilization of the concrete pressure zone compared to the variant from Figure 3b in a bending component designed for the ultimate limit state. The effects investigated for the general case are explained in detail in Reference [47] and can be summarized as "lying on the safe side" since both variants do not have an intersection with the 5% or 95% quantiles of the probabilistically calculated design parameters. In Reference [47], the variant from Figure 3c was defined as the preferred variant.

3.4.2. Partial Safety Factor for Carbon Textiles

Safety factors are a crucial topic within the framework of a semi-probabilistic safety concept, which is common in the construction industry. In 2014, a proposal was already presented for the determination of partial safety factors for carbon textiles and for the characteristic and design values of the tensile strength, see Reference [25]. The distribution and the mean value of the strength were determined on the basis of 50 uniaxial tensile tests. Assuming a normally distributed quantity, a standard deviation and the 1.5% quantile were calculated, considering the low data basis for statistical purposes. The characteristic nominal strength for the investigated carbon textile acc. to [25] was specified to 1550 N/mm$^2$.

In Reference [53], a comparative calculation was presented using EC0 [54] on the basis of a representative bending component. The general safety level for buildings and components required by the Eurocode is verified by calibration on reinforced concrete. First, a steel reinforced concrete slab is dimensioned using the usually applied semi-probabilistic method and the required reinforcing steel reinforcement is determined. Subsequently, the distributions of the acting and resisting moment are calculated and compared using a Monte Carlo simulation under the assumption of scattering actions (dead weight and payload) and resistances (reinforcement tensile strength), so that a failure probability can be determined first for the steel reinforced concrete slab. Afterwards, the same procedure is carried out in several iteration steps for the textile reinforced concrete slab: the iterated parameter equals the partial safety factor for the textile and the iteration target equals the failure probability of the steel reinforced concrete slab, which was considered safe. With the aid of this procedure, the partial safety factor for textiles $\gamma_t = 1.2$ was introduced into the first building authority approval [25]. More recent investigations with further calculations [55], however, showed that a lower value with $\gamma_t = 1.1$ also is reasonable. These partial safety factors do not take into account reductions in regard of durability, temperature and permanent load. In Reference [53], all influences were combined to a general, and therefore significantly higher, safety factor. However, since these reductions differ for each type of material, it is recommended to adjust the corresponding characteristic value. The topic is still under research in textile concrete research, e.g., in the $C^3$ project [56].

## 4. Practical Applications

Even if there are still points open for discussion, TRC has already been used for several renovation measures. An overview on the different fields of application are given e.g., in References [22,24,57], additional examples are presented for example in References [58–60]. TRC is used for the strengthening for static or earthquake loading, but also to improve the usage properties, e.g., limiting deflections, repairing cracks, and sealing or repairing concrete surfaces. In the field of flexural strengthening, examples of applications can be found both in conventional building construction and in the field of monument protection.

An example of the first group is the renovation of a residential and commercial building in Prague (Czech Republic, 2009/2010) [59]. Ceiling slabs had to be strengthened in that project. The RC flat slabs were supported by brickwork at the edges and by point supports inside the building (column grid 12.8 m × 13.1 m). In some areas there were considerable deflections of up to 15 cm. In addition, there were problems with the flexural load-bearing capacity and the punching safety. TRC was used in the interior fields to increase the bending load-bearing capacity and the ceiling's stiffness. Up to 4 layers of carbon textile were required. After preparing the surface (roughening, pre-wetting), the first layer of fine-grained concrete (3 mm thick) was sprayed on. The carbon layer was subsequently embedded. These steps were repeated until the required number of layers was reached. The maximum thickness of the TRC layer was 2 cm. A total of approx. 3000 m$^2$ of carbon textile was processed.

In the field of monument protection, TRC has already been used for the renovation of several RC shell structures, where the low additional weight and the flexibility of the textile fabric are particularly advantageous. A further early application was the renovation of a hypar shell above a lecture hall at the Schweinfurt University of Applied Sciences (Germany, 2006), e.g., [61,62]. The shell, which was only 8 cm thick in the middle area, measures 38 m × 39 m. In the area of the cantilevered high points, considerable deformations occurred due to stress exceedances over the column supports. The strengthening was carried out with 3 layers of carbon textile (Figure 6a). A total of approx. 450 m$^2$ of carbon textile was processed. A second RC hypar shell was retrofitted with carbon reinforced concrete in Templin (Germany) in 2016. The refurbishment of a hypar shell in Magdeburg (also in Germany, Figure 6b) is currently being planned [63]. The tests to obtain an approval for individual cases based on Reference [25] were successfully completed. The strengthening with carbon textiles is scheduled for 2019.

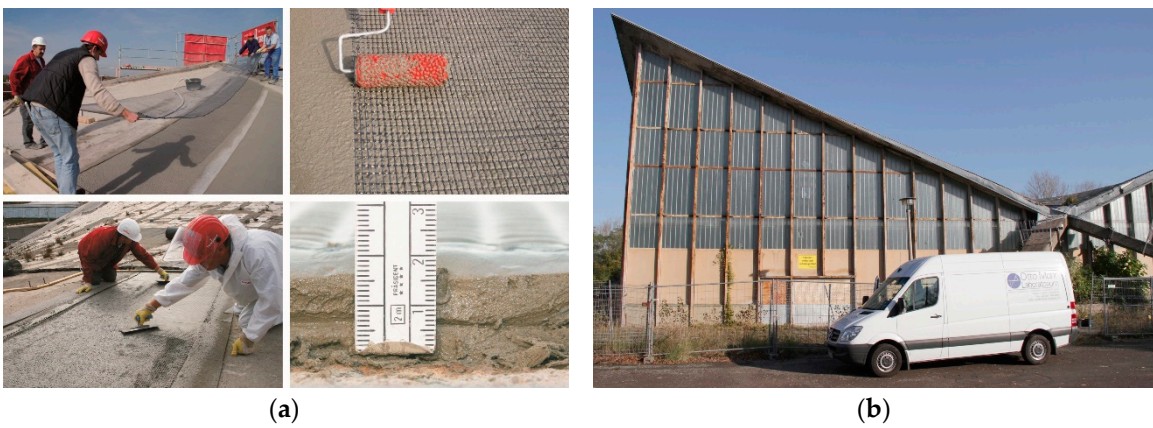

**Figure 6.** Textile reinforced concrete (TRC) retrofitting of hypar shells; (**a**) application of TRC strengthening at a hypar shell in Schweinfurt (photos: Ulrich van Stipriaan and Silvio Weiland, reproduced with permission from [59], Ernst & Sohn, 2015), (**b**) hypar shell in Magdeburg before restoration (photo: Heiko Wachtel).

Until today, mostly structural engineering components have been retrofitted with textile reinforced concrete. Yet, there is also a considerable need for renovation of bridges. TRC has already been tested on concrete arch bridges (non-reinforced concrete), bridge piers, bridge caps or, as an additional concrete layer, on bridge decks (compression zone), see e.g., References [58,64–66].

## 5. Outlook on Current Research and Summary

Bending tests were also carried out within the BMBF-funded project 'C$^3$ – Carbon Concrete Composite' [56], subprojects C3-V1.2 [67] and C3-V2.7 [68] in at TU Dresden. The aim of research and development within 'C$^{3'}$ is to establish carbon reinforced concrete in construction practice. Both the carbon reinforcements and the concretes used were further developed in order to improve the properties of the composite (durability, manufacture, anchoring, etc.). Load bearing mechanisms as well as design models are to be investigated and modified respectively in greater depth. In comparison to the textiles used in earlier bending tests, the now used ones are significantly stiffer and show higher tensile strengths. The main aim of the test program was therefore to demonstrate that the calculation model presented in Section 3 for strengthened RC slabs can also be applied to the new textiles. The tests were carried out on plates with the dimensions $3.3 \times 0.5 \times 0.12$ m$^3$ mainly as four-point bending tests with variable distances of the load introduction points. In addition to six reference plates, a total of 18 strengthened plates were tested. For strengthening, textiles from solidian GmbH (GRID Q85/85, $a_t = 85$ mm$^2$/m) and from V.Fraas Solution in Textiles GmbH (SITGrid 40, $a_t = 141$ mm$^2$/m) were used. An evaluation of the experiments is currently taking place. Therefore, only the basic phenomena, observed during the experiments, are mentioned here. For an in-depth scientific analysis, please await future publications.

Depending on the configuration of the textile and the concrete base, the strengthened plates achieved 2–6 times the loads compared to the reference ones. In principle, the load-bearing behavior was in accordance with the effects described and observed earlier. The unstrengthened reference plates with low reinforcement level showed a tensile failure of the steel reinforcement; at higher reinforcement levels, a concrete compression failure occurred. The predominant type of failure of the strengthened slabs was a tensile failure of the textile reinforcement. At very high TRC strengthening levels, the formation of a critical crack could be observed, which led—after constricting the pressure zone—ultimately to a compression failure of the concrete in the pressure zone. This type of failure occurred at loads at which the range of minimum shear force resistance of the concrete base (calculated acc. to References [41,48,49]) had been nearly reached or already exceeded. In addition, in some configurations, splitting, or delamination in the plane of the textile reinforcement was observed,

starting with a crack. This tendency was observed to an increasing extent with the reduction of the distance between the two individual loads. The exact reasons for the change of failure mode, however, have still to be analyzed. One assumption might be the curvature in connection with the stiffness of the textile. The observed failure types and forms lead to the conclusion that in addition to the desired flexural strength failure, other failure types can also become decisive due to the higher stiffness of the new generation of textile reinforcements. This must be taken into account in the planning and execution of strengthening measurements. In addition, proofs must be developed in order to reliably calculate these types of failure.

In summary, the article outlines the potential of flexural strengthening with TRC. A calculation model was presented to dimension a TRC strengthening. The assumptions underlying the model were explained and discussed. Critical points were identified. Thus, for the application of the model, for further research and development of TRC construction and for the practical application of TRC, the type of the textile reinforcement, the associated stress–strain curve, the bond behavior, and a preloading of the basic component must be taken into account. Hence, further research will be necessary with regard to the change of failure modes.

**Author Contributions:** Conceptualization, M.C.; Formal analysis, R.Z., E.M. and A.S.; Funding acquisition, M.C.; Investigation, R.Z., E.M., T.S.-P. and A.S.; Methodology, R.Z.; Project administration, M.C.; Supervision, M.C.; Validation, S.S., R.Z. and M.C.; Visualization, S.S., E.M., T.S.-P. and A.S.; Writing – original draft, S.S., E.M., T.S.-P. and A.S.; Writing – review & editing, S.S., R.Z. and M.C.

**Funding:** The findings presented in this paper are the result of several projects carried out over the last 20 years. The German Research Foundation (DFG) should be mentioned first and foremost. In addition to many other fundamental investigations on TRC, more than 70 flexural tests on large-format plates were examined during the DFG-CRC 528 'Textile Reinforcement for Structural Strengthening and Repair' (project number 5483454, funding period: 1999–2011), e.g., [69,70]. Within the frame of the test programme for the general building approval [25] (funded by TUDALIT e.V. [71]), further approx. 50 plate tests were carried out together with the Institute of Concrete Structures of RWTH Aachen University; in addition, the design model was improved and prepared for application in engineering offices, see [26,45,46,55,72]. Furthermore, at our institute, numerous bending tests were carried out for approvals in several individual cases for various construction measures (different funding organizations/companies, different years). Since 2014, the project consortium 'C$^3$—Carbon Concrete Composites' [56] with circa 170 German partners has been funded by the German Federal Ministry of Education and Research. Flexural tests on strengthened slabs are mainly carried out in the projects C3-V1.2 'Verification and testing concepts for standards and approvals' [67] (funding period: 01.2016–04.2018, grant number: 03ZZ0312A) and C3-V2.7 'Development of a general plan to strengthen an existing concrete structure with carbon reinforced concrete' [68] (funding period: 05.2017–04.2020, grant number: 03ZZ0327A).

**Acknowledgments:** The results presented here were generated in different projects over the past 20 years. This research is only possible with the support of our colleagues in the Otto Mohr Laboratory, where most of the experiments were carried out. We would like to express our sincere thanks to them! Further thanks go to the colleagues and partners in the various completed and ongoing research projects; above all to research group 2 of the Institute of Concrete Structures of TU Dresden and the partners of the numerous C$^3$ projects. We would also like to thank all the companies that have provided us with research material free of charge. Thanks also to Dajana Musiol for reviewing the English paper.

**Conflicts of Interest:** The authors declare no conflict of interest.

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
