# Peer review of "Flexural Strengthening of RC Structures with TRC—Experimental Observations, Design Approach and Application"

_applsci, doi:10.3390/app9071322_

Reviewer 1 Report

The manuscript provides a brief overview of the basic structural behaviour of RC members strengthened in flexure using TRC. Moreover a design model is presented, its assumptions and limits are analysed and discussed, whereas selected case studies are finally presented. The overview of the presented work is based on research conducted mainly in Germany, and the literature review needs to be enriched. Overall, the review presented is interesting and could be published in the journal; however, the authors should revise their manuscript to address a number of points (see pdf attached) before the paper can be accepted for publication.

Author Response

Dear Reviewer,

please find our answers in the pdf file.

Best regards.

Reviewer 2 Report

The paper presents an interesting overview of the application of TRC for strengthening RC structures subjected to bending. Results are presented in a clear and effective manner, with an adequate level of detail.

Author Response

(The authors gave the same response as above.)

Reviewer 3 Report

Journal

Applied   Sciences 

Article Title

TRC Strengthening for RC Structures   Subjected to Bending – Experimental Observations, Design Approach and Application

Article ID

applsci-460441

Type

Reviewer’s   Comments

Date

Tuesday, February 26, 2019

General Comments

The article treats a   very interesting and crucial issue. It is   suggested that this manuscript, based on its current form be entitled as a review article on the TRC   Strengthening. The provided material may be improved   by adding other studies in order to complete an extensive literature review. At   the present form, the article is not sufficiently adequate to be considered   as a research article thence it is required by the authors an extra and serious work for enhancing it. Some   suggestions may be found in the   following comments.

The authors have investigated the possibility of   retrofitting the RC elements subjected to bending, from experimental to   design approaches. A clear distinction between their contribution and the   literature review is not achieved. It   is essential to be present such aspect, in order to highlight the contribution   of this article in state of the art.

The manuscript fulfills the scope of the journal.   In this regard, the paper deserves the attention of the editorial board.   However, for the sake of clarity, the submitted manuscript may be considered   for publication only after the authors raise the following comments and carry   out a major revision of their   manuscript.

Specific Comments

·           Section 2 of   the manuscript does not provide a comprehensive literature review of the   topic. Instead of the research history at TU Dresden the reader would be   interested to have a literature review and the advancement of the research,   in terms of innovative approaches from the TU Dresden. It is too long and with   many trivial statements.

·           Section 3 is quite long. It would be more beneficial   to be shorter and more concise as far   as no clear contribution from the   authors is present. The authors say: “The   material properties strength and elongation at failure of the textiles are distributed stochastically”. The sentence is entirely out of context, or the authors have not provided adequate justification on the material   properties, on how do they take them.

·           The presence of section 4 is more like an addition   not in the right place and not correctly implemented. A review of it would be   mandatory.

·           Section 5, the one that was expected to be the   most important one is inexplicable vague. The current research is not clear. The   set-up of the experiment is missing, and it cannot be judged! The authors say: “The   experiments are currently being evaluated.   Therefore, only the basic phenomena, observed during the experiments, should be mentioned here.” In my opinion,   the chosen word “should be” is wrong and, scientifically it is unacceptable   to provide results not in the correct   way.

·           Regarding the conclusions, the authors have not pointed out, both clearly and not clearly their scopes and contributions.

English

·           English is   not perfect and requires professional editing. In some cases, the German   was used instead. The manuscript should   be free from errors.

Final Statement

The above remarks may help in the improvement and   completeness of paper in terms of,   consistency, clarity, and accuracy of   scientific results, by means of careful addressing from the authors. All things considered, the paper is considerable for publication in the “Applied   Science” journal providing the authors to   account for the comments raised.

Author Response

(The authors gave the same response as above.)

Reviewer 4 Report

General comments:

This is a very interested research paper of a research project about the structural performance of textile reinforced concrete as a structural strengthening system of existing concrete structures. The collaboration between the existing concrete structure and the new applied layer of textile reinforced concrete is a subject of high engineering interest. The structural analysis and the respective analytical models of the stress distribution between the existing and the new structural components seem to be very interesting. In any case a detailed numerical model using Finite Element analysis Model is missing (or not referred) of this research project. Such an analysis is necessary for comparison and evaluation of the analytical and experimental tests of this composite strengthening system.

 Detailed comments:

1.      Syntax errors have been found frequently in a few sentences of the paper which need to be rearranged (i.e. chapter 2 “At a deflection of 8.5mm resp…………………strength”).

 2.      In chapter 1 the sentence “Reinforced concrete has been the world’s most widely used building material…." should be replaced by the phrase (Reinforced concrete has been the world’s most widely used composite building material….”.

 3.      The headings in paragraphs 3.3.2 and 3.3.3 are very general and need to be more specific.

 The main critical points that already noted during this research study should be included in the chapter 6  “6. Summary and Conclusions”.

Author Response

Dear Reviewer,

please find our answers in the pdf file.

Best regards.

Round  2

Reviewer 1 Report

The authors revised most of the requested comments, and the manuscript has been significantly

improved. Therefore it is suggested to be accepted for publication to the journal.

Reviewer 3 Report

The authors have addressed all the issues and have carefully improved their work. After the revision of the authors on the original manuscript, the paper is recommended to be published in the “Applied Science” journal.